# Virtual Reality (VR) Technology for Treatment of Mental Health Problems during COVID-19: A Systematic Review

**DOI:** 10.3390/ijerph19095389

**Published:** 2022-04-28

**Authors:** Muhammad Hizri Hatta, Hatta Sidi, Chong Siew Koon, Nur Aishah Che Roos, Shalisah Sharip, Farah Deena Abdul Samad, Ong Wan Xi, Srijit Das, Suriati Mohamed Saini

**Affiliations:** 1Department of Psychiatry, Universiti Kebangsaan Malaysia Medical Centre, Cheras, Kuala Lumpur 56000, Malaysia; p113778@siswa.ukm.edu.my (M.H.H.); hattasidi@hotmail.com (H.S.); shalisah@ukm.edu.my (S.S.); farahdeena.as@gmail.com (F.D.A.S.); 2Department of Psychiatry, Hospital Sultanah Nur Zahirah (HSNZ), Kuala Terengganu 20400, Malaysia; chong_siewkoon@yahoo.com (C.S.K.); drisaong@gmail.com (O.W.X.); 3Faculty of Medicine and Defence Health, National Defence University of Malaysia, Sungai Besi, Kuala Lumpur 57000, Malaysia; nuraishah@upnm.edu.my; 4Department of Human & Clinical Anatomy, College of Medicine & Health Sciences, Sultan Qaboos University, Al-Khoud, Muscat 123, Oman; s.das@squ.edu.om

**Keywords:** virtual reality, tool, psychological, emotional, mental, health, problems, pandemic, review

## Abstract

There was a surge in psychological distress and emotional burnout during the COVID-19 pandemic. Virtual reality (VR) is helpful as a psychological intervention whilst maintaining physical or social distancing. The present systematic review assessed the role of VR as a psychological intervention tool for mental health problems during the COVID-19 pandemic. We conducted a systematic review that followed the Preferred Reporting Items for Systematic Reviews and Meta-Analysis (PRISMA) guidelines. This study used the search-related terms: (Virtual reality OR simulated-3D-environment OR VR) AND (covid! or corona!) AND (mental* OR psychologic* OR well* OR health*) AND (intervention) on six databases, i.e., MEDLINE, PsycINFO, Ovid Medline, EMBASE, ACM digital library, and Cochrane Central Register of Controlled Trials (CENTRAL) from the inception date until 23 June 2021. We finally included four studies in the systematic review out of the 379 references imported for screening. These studies reveal that VR is beneficial as a psychological tool for intervention in individuals with mental health problems. Immersed in the telepresence, interacting in a 3-D format compared to a 2-D layout, having a sense of enjoyment and engagement, activating an affective-motivational state, “escaping” to a virtual from the real world are pivotal faucets of VR as a psychological tool for intervention.

## 1. Introduction

The coronavirus disease 2019 (COVID-19), caused by the inexplicable SARS-CoV-2 virus, was first identified in Wuhan, China, in December 2019, and gradually spread to the other parts of the world [1]. The disease was found to be extremely contagious and resulted in high mortality. Since then, the virus has swiftly spread everywhere in China and other parts of the globe. Subsequently, the World Health Organization (WHO) declared this infectious condition a global pandemic in March 2020 [2]. The devastating pandemic resulted in the loss of livelihoods due to prolonged lockdowns, and this had a negative effect on the global economy [2]. There was a mass spread of disease in early March 2020, and then, it became worse [3]. Many countries enforced restrictions or movement control orders (MCOs) from March 2020 to June 2020, and until 2022 [4].

The MCO imposed a ban on the public from socializing and taking part in any events or gatherings. These included cultural, educational, social, and religious activities such as praying and gathering at religious facilities [5]. Sports activities, such as walking outside the house, cycling, jogging, work-related (for non-essential sectors), and scholarly activities, such as face-to-face classes, were restricted and strict social distancing methods were imposed [6,7]. There was much psychological and emotional distress among individuals. Studies also observed the change in emotions and factors associated with emotional distress immediately as well as following the onset of the pandemic [8]. 

Depression, grief, anxiety, anger, irritability, and emotional burnout throughout the MCO period was observed during the pandemic [6]. The public, patients, healthcare workers, and medical personnel were all at risk of contracting a viral infection on a regular basis. This situation predisposed them to a higher risk of major mental illnesses, which included depression, anxiety, post-traumatic, and sleep disorders during the challenging time of the pandemic [9,10,11,12,13,14]. A study which investigated the MCO and COVID-19 pandemic’s psychological impact, reported a high prevalence of psychological morbidity in a cohort of university students during this pandemic [15]. Because of the limitation of face-to-face psychological interventions, there was a need for different apps for digital reformation to treat individuals with mental health problems. 

COVID-19 is a very contagious disease and it spreads rapidly. There was a directive from the administration to remain at home and avoid going out. An individual could not easily access natural beauty and attractions for any purpose, including recreation. Because virtual environments can generate the required visual, cognitive, and social links to connect the individual to the living place, they can evoke favourable psychological responses [16]. Various research studies have found that VR-based therapies may be beneficial for various mental health disorders [17,18]. VR has been employed as a distraction device to overcome patients’ stress [19,20,21]. 

To date, there has been a paucity of information on how psychological interventions such as digital reformation, i.e., the use of apps to treat patients with COVID-19-related mental-health problems, will be helpful. The lack of published literature has added to the knowledge gap in this specialty area. The majority of studies on psychological intervention were conducted online, without an interactive session to help patients deal with their problems. To the best of our knowledge, there are no studies that assessed the psychological intervention provided by digital interactive navigation during the COVID-19 pandemic. Many literature reviews on major psychiatric illnesses, such as major depressive illness, anxiety disorders, and other stress disorders, did not include the role of VR. It is pivotal to perform a systematic review to provide a comprehensive and up-to-date review of the role of VR as a psychological intervention for mental health problems during the COVID-19 pandemic. 

## 2. Material and Methods

This systematic review conformed to the Preferred Reporting Items for Systematic Reviews and Meta-Analysis (PRISMA) guidelines [22,23] and was registered in the International Prospective Register of Systematic Reviews (PROSPERO) under registration number CRD42021265380.

### 2.1. Search Strategy

Search strings were developed and conducted across the following electronic databases: PsycINFO, Ovid Medline, Embase, ACM digital library, and Cochrane Central Register of Controlled Trials (CENTRAL) from December 2019. We conducted the last search on 23 June 2021. The search-related terms incorporating the Boolean and the truncation were as follows: (Virtual reality OR simulated-3D-environment OR VR) AND (covid! or corona!) AND (mental* OR psychologic* OR well* OR health*) AND (intervention).

We restricted the searches to human studies and publications in the English language. No geographical restriction was applied.

### 2.2. Eligibility Criteria

A study was eligible for inclusion if it met the following criteria. Regarding the study design, only primary studies such as case–control studies, cross-sectional studies, cohort studies, and randomized controlled trials were considered.

Regarding the respondents, the criteria included adult subjects with or without underlying psychological distress, including depression, stress, anxiety, and psychosis secondary to COVID-19, a relative affected by COVID-19, or being infected by COVID-19. No gender, race, or ethnicity restrictions were applied. The criteria for interventions/exposure included psychological interventions, e.g., counselling, family-based intervention, psychotherapy, behavioural therapy, and positive activity intervention (PAI) delivered using VR in an inpatient or outpatient setting. The comparator includes any conventionally delivered psychological interventions. The outcomes included improvements in psychological wellbeing. The outcomes were measured by subjective report or objectively assessed using tools such as the Depression Anxiety Stress Scale (DASS), General Health Questionnaire (GHQ), and Beck Depression Inventory (BDI). 

The exclusion criteria included (i) studies conducted on children and adolescents; (ii) intervention in the form of other immersive technology, e.g., augmented reality (AR); (iii) abstracts, conference proceedings, case reports, case series, and reviews; and (iv) studies that could not report the outcome of interest. 

### 2.3. Selection Process

We exported all search results into a reference management software, Covidence [24]. The program automatically takes out all duplicate studies. Two independent reviewers (MH and CSK) screened titles and abstracts for eligibility. The full texts of the eligible records were then obtained, and screened for eligibility according to the PICOS framework with the help of two independent reviewers. To achieve consensus, we resolved any doubts or conflicts by discussion with a third reviewer (H.S.). 

### 2.4. Methodological Quality

We assessed the quality of each of the included studies according to the McMaster Critical Appraisal Tool for quantitative studies. We also added questions on the randomisation for quality assessment of randomised-controlled trials (Appendix A). Each question was rated with a “yes,” “no,” “not addressed,” or “not applicable.” Every “yes” answer was given a score of 1 point, and the overall score differed depending on the study design. The intervention category of the Australian National Health and Medical Research Council’s (NHMRC) evidence hierarchy was also used to determine the level of evidence in the included studies. 

### 2.5. Selection Process

We exported all search results to reference management software, Covidence (Covidence systematic review software, Veritas Health Innovation, Melbourne, Australia; available at www.covidence.org, accessed on 28 July 2021). The program automatically removes duplicate studies upon import. Titles and abstracts were screened, and then full texts were analysed for eligibility following the inclusion criteria by two independent reviewers (M.H., C.S.K. and F.D.). Any doubts or conflicts were resolved by discussion between the two reviewers, and we thus reached a consensus. 

### 2.6. Data Extraction

Three independent reviewers (M.H., C.S.K. and F.D.) performed a data extraction standardized form created in Microsoft Excel (Microsoft Corporation, Redmond, WA, USA). The first author, year of publication, number of participants, age, country of origin, study setting, exposure, comparator, length of follow-up, and outcome were extracted from each included study. We contacted the authors via email to obtain additional information if necessary. Any discrepancies in the extracted data were resolved by a third reviewer (H.S.). 

### 2.7. Methodological Quality

We assessed the quality of each of the included studies according to the McMaster Critical Appraisal Tool for quantitative studies [25]. A revised version was produced by removing the domain-assessing interventions, as they were not relevant to this review. Questions on randomisation were also added for the quality assessment of randomised-controlled trials (Appendix A). Each question was rated “yes,” “no,” “not addressed,” or “not applicable.” Every “yes” answer was given a score of 1 point, and the overall score differed depending on the study design. The intervention category of the Australian National Health and Medical Research Council’s (NHMRC) evidence hierarchy [26] was also used to determine the level of evidence of the included studies. 

### 2.8. Data Synthesis

The study quality and characteristics of interest were tabulated and narratively described.

## 3. Results

### Study Characteristics

Out of the 379 references imported for screening, 29 duplicates were removed. A total of 350 studies were screened against titles and abstracts. Subsequently, 302 studies were excluded, and among the 48 studies assessed for full-text eligibility, 44 studies were excluded because of the following reasons, i.e., wrong study design (21) (e.g., case reports), wrong intervention (6) (e.g., augmented reality (AR), face-to-face consultation), wrong outcomes (6) (e.g., education satisfaction), wrong patient population (5) (e.g., healthy or not related to mental health), wrong setting (2) (outside the time frame of the COVID-19 pandemic), no outcome findings (1) (e.g., research protocol with no outcome results), trials registration (1), wrong indication (1), and commentators (1). A total of four studies were finally included (Figure 1). We summarized the key results of the study characteristics, which include information about the intervention, its type, duration, setting, and effectiveness in Table 1. 

## 4. Discussion

To the best of our knowledge, this study was the first review to explore the role of VR as a psychological intervention for individuals with mental health problems. We included important studies by Yang et al. (2021), Siani et al. (2021), Waller et al. (2021), and Kolbe et al. (2021) in our final search criteria based on the systematic review study [27,28,29,30]. We could not synthesize the data for a meta-analysis because of the vast differences (heterogeneity) between the studies, i.e., different populations, assessment tools, different outcomes, and the duration of studies. 

Regarding the four studies, their quality was fair to good. The studies by Yang et al. (2021), Siani et al. (2021), Waller et al. and Kolbe et al. (2021) [27,28,29,30] showed fair (both cross-sectional study and comparative study without concurrent controls, type III) and good (RCT, type II) quality, respectively. 

### 4.1. The Role of 360° Virtual Tour in Psychological Stress Reduction

This study constructed a four-level model to examine how a 360° virtual tour can reduce people’s psychological stress on two types of presence, i.e., the sense of existence or telepresence, and affective-motivational, referring to the emotions and rewards during the extraordinary period of the COVID-19 pandemic. Yang et al. (2021) [30] employed the partial least squares (PLS) analysis to test the moderating effect of a person’s involvement among two hundred and thirty-five respondents. This study showed that telepresence (TP) had a more significant impact on generating affective-motivational states (AMS) than the sense of presence. Among the factors, enjoyment [31,32,33] showed the highest effect on satisfaction with the 360° virtual tour experience and stress reduction. The engagement of an experience and AMS moderated the impact of TP on satisfaction with the 360° virtual tour adventure. Yang et al. (2021) [30] reported VR research outcomes by differentiating between the concepts of ‘sense of presence’ and ‘telepresence’ and demonstrating how VR technology can influence an individual’s psychological and mental well-being. 

For practical implications, when designing a 360° virtual tour, many software developers (SD) emphasised the experience of a ‘real’ situation in the VR platform. However, users have a sense of ‘being there’ instead of being in the ‘real place.’ Just like the makers of Marvel/Warner Bros. cinematics, the SD has to plan to produce a sense of ‘being there’ among the viewers to encourage viewers to watch their movies. Respondents to a tourist destination are looking for the importance of ‘being there’, as more than a ‘real’ presence in their memory destination. As they discovered, they enjoyed 360° content more as an ‘ideal place’ destination. For example, they were watching a turquoise sky and a vibrant rainbow. These virtual tour experiences they could never see frequently in the ‘real place’ than in the ‘ideal place’ are gratifying experiences. The enjoyment of physical activity in immersive VR is another way to benefit from digital health creativity [34]. They retained their experiences in the post-processed memory effects. The AMS is considered the response to the 360° virtual tour content. When exposed to the ‘ ideal place’, the respondents could have more gratification and involvement (based on audio, screen clarity, and even the sense of smell of the real places) as part of the AMS process.

The SD has to design the 360° virtual tour experience as an immersive experience as much as possible so that participants can imagine themselves as tourists. VR developers could co-design with the respondents to integrate their interests and improve the flow elements of the 360° virtual tour content. It is time to recommend reducing psychological stress during and after the COVID-19 pandemic. The 360° virtual tour also offers an experience for senior citizens with mobility issues [30]. For instance, senior citizens who cannot take a long-distance trip may benefit from the virtual tour and help promote an old person’s mental well-being.

### 4.2. VR Video Games as a Form of Recreational Use during the Lockdown 

Siani et al. [28] aimed to evaluate the effects of VR activities on users under lockdown because of the COVID-19 pandemic. They investigated the recreational use of VR during the lockdown period and gathered users’ opinions on its impact on their physical and mental healths. The researchers employed non-parametric tests to evaluate the statistical significance of the responses provided by the 646 participants. The survey results showed that VR use significantly increased during the lockdown period for most participants, who expressed overwhelmingly positive opinions on the impact of VR activities on their mental and physical wellbeing. Interestingly, the self-reported intensity of physical activity was considerably more strenuous in VR users than in console users. 

Video games were the most popular VR activity [28], followed by physical fitness, social media (SM), videos, and meditation. Most of the respondents spent 1–4 h per day playing VR video games and up to 2 h/day using VR for physical fitness. Most of the respondents used up between 1 to 4 h a day for playing VR video games and a VR for physical fitness up to 2 h/day. Interestingly, VR video games require substantial physical effort. Less than 20% of those polled spent more than an hour/day using VR for video, social media, or meditation activities. Even with brief VR sessions, such as a 10 min 3D 360° video daily for a week, regular engagement could relieve lockdown-induced boredom and anxiety and subsequently foster a positive psychological state. Regardless of gender and age, VR use increased during lockdown and is associated with a positive opinion on the usefulness of VR to keep respondents busy and improve their psychological and physical well-being. Siani et al. (2021) [28] found a variation in self-reported workout intensity between respondents using gaming consoles and VR headgear. They linked the latter to a significantly more intense exercise than the former. These findings imply VR may be a more effective device in promoting training/activity than traditional consoles. 

Siani et al. (2021) [28] provided novel insights into how the recreational use of VR can successfully alleviate the negative impact of lockdown periods on the population’s mental and physical wellbeing. VR activities help users keep themselves occupied and physically active under the restrictions imposed by the lockdown. VR headsets have become a mainstream entertainment device in many households, because of their increasingly affordable price and high-tech accessibility. Therefore, researchers, policymakers, and healthcare workers should consider designing and implementing intervention strategies as potential aids in public health to mitigate the negative consequences of prolonged lockdown periods. Providing the population with the means to engage in VR activities to keep them occupied and physically fit could be a promising strategy to minimise the decline in mental and physical well-being. Therefore, this digital health intervention might ease the current pressure on healthcare and medical workers.

Given the current uncertainty regarding the duration and course of the pandemic, as well as the possibility of intermittent lockdowns in the upcoming years, the outcome of this study could have a significant impact on the development and deployment of VR-based strategies which aim to help the population cope with prolonged social distancing, particularly with regard to vulnerable individuals.

### 4.3. Meditative Effect of 3D vs. 2D Format 

This strategy by Waller et al. (2021) [27] includes meditation instruction. In VR, a person feels the physical presence of an instructor, although this may be lessened when the interventions are employed remotely. This situation might affect one’s meditative experiences. The use of head-mounted displays (HMD) to demonstrate video-recorded instruction (VRI) may enhance one’s sense of psychological presence with the instructor as compared to the presentation via a regular flatscreen (e.g., laptop) monitor. Waller et al. (2021) [27] studied the role of meditating by employing VR in a 360° video of perception in the presence of an instructor. 

The researchers evaluated a didactic, trauma-informed care approach to instruction in mindfulness meditation by comparing meditative responses to an instructor-guided meditation. It is delivered face-to-face versus by pre-recorded 360° videos viewed either on a standard flatscreen monitor (2D format) or via HMD (virtual reality (VR) headset; 3D arrangement). All 82 respondents were recruited from a university introductory course. They were asked to experience a 360° video-guided meditation via HMD, i.e., VR conditions in a 3D format. They were also randomly assigned to employ the same meditation either via scripted face-to-face instruction (in vivo (IV) format) or when viewed on a standard laptop display (non-VR condition, 2D design). Researchers documented respondents’ maintenance of focused breathing attention using the meditation breath attention scores (MBAS) throughout each meditation session. Meditating in VR (3D format) was associated with a heightened experience of amazement. Compared to face-to-face instruction (in vivo format), VR meditation was rated as less embarrassing, less entertaining, and extra tiring. When compared to the 2D format, VR meditations were associated with more significant relaxation experiences. The latter was associated with less distractibility from the practice of breathing and less exhaustion. Based on MBAS, there were no significant differences found between VR and non-VR meditation in the respondents’ concentration. Baseline post-traumatic stress symptoms were risk factors for experiencing distress while meditating in either a VR/3D or non-VR/2D instructional format. Approximately half preferred the VR format and recorded 360° video instruction in meditation viewed with an HMD (i.e., VR/3D form), which appears to offer some experiential advantage over instructions given in 2D format. VR meditation may provide a safe—and even preferred—alternative to face-to-face teaching meditation.

### 4.4. The VR Program Implemented in COVID-19 Recovery Unit (CRU)

Based on Kolbe et al. (2021) [29], VR has seen extended use in acute healthcare and clinical settings in recent years. Many mental health issues were present, including psychoneurological sequelae, adjustment disorder, depression, and stress [4]. The study by Kolbe et al. (2021) [29] suggested that the innovative use of VR is one modality of a part of rehabilitative CRU due to its ability to alleviate distress. Kolbe et al. (2021) [29] discovered positive patient fulfilment and a perceived benefit of VR on patients and medical staff in the CRU, as well as demonstrated the feasibility in logistical functioning and VR content delivery. During the COVID-19 surge in New York City in 2020, the CRU in a large teaching hospital included respondents with three categories of interactive experience, i.e., guided meditation, exploration of natural environments, and cognitive stimulation games.

The participants were surveyed regarding satisfaction and whether they perceived and benefited from the sessions. The study found that 13 patients and 11 staff reported median patient satisfaction scores of 9 out of 10. Ten patients showed to be “extremely satisfied,” with a median staff satisfaction score of 10. All patients responded “yes” to endorsing the VR intervention. A total of 92.3% of patients approved the perceived enhancement of their therapy. A total of 100% of staff agreed to recommend the intervention to others. They also agreed to the role of VR as a perceived augmentation of their well-being. Kolbe et al. (2021) [29] reported that both patients and healthcare providers were extremely satisfied with the perceived benefit of the VR program implemented on a CRU. Participants stated that the use of VR was valuable and beneficial in coping with the sense of isolation and loneliness. They also stated other reasons, as depicted in Figure 2. 

VR could be implemented within medical care for COVID-19 patients as an ideal rehabilitation model. Using VR was also logistically and operationally feasible in the CRU. Future work to compare the positive benefits of VR to a novel neuropsychological intervention and rehabilitation is needed.

In summary, Kolbe et al. (2021) [29] discovered that a VR program implemented on a CRU is extremely satisfying for both patients and staff, with perceived benefits for improving patient treatment and healthcare staff well-being. 

The acceptance of VR use in the CRU has not only made patients more interested in the ongoing use but has also generated staff interest in the expansion of VR use in inpatient therapeutic settings. The next challenge would conduct a more extensive prospective study to assess whether exposure to the VR content could promise better outcomes, such as mood elevation, anxiety control, sleep regulation, pain management, and a reduction in feelings of isolation. 

### 4.5. Shared Elements for VR in the Psychological Intervention for a Patient with Mental Health Problems

The studies by Yang et al. (2021), Siani et al. (2021), Kolbe et al. (2021), and Waller et al. (2021) [27,28,29,30] shared similar or related elements in the role of VR as a psychological intervention for a patient with mental health problems. The studies concluded that VR had a beneficial effect as a tool for digital intervention among distressed individuals. In this systematic review, we found that immersion plays a crucial role in the mediation of therapeutic benefits [29]. “Immersion” refers to the perception of being physically present in a non-physical world [35]. The encircling environment experienced by the user-generated perception in the VR system will interact with stimuli such as images and sounds that provide an engaging atmosphere. The interactive experience between the users and the digital platform in a 3-D setting (VR design) is preferable to a 2-D scene (i.e., flat screen). The VR platform as a digital health intervention is also helpful for users who want a sense of self-empowerment (ability to perform the intervention at their convenience or leisure) vs. predetermined regimented therapy (performing the intervention at a given specific time and by a particular instructor or therapist). This freedom and flexibility in terms of time and space are crucial for the use of VR as a tool for in-demand psychological intervention, especially among distressed individuals who are constrained by busy schedules and the need to attend a face-to-face consultation.

Our review also found that engaging in the VR scenario and interacting with the stimuli during the immersion process is pivotal to the beneficial effects of the psychological intervention [36]. Activities such as gaming, guided medication, virtual tours, and probably gauging the biological response could be helpful in the process of the intervention. However, there is a knowledge gap between how long a person should engage in the VR platform (i.e., 30 min or more) and what kind of physical activity in the VR system would provide benefits (gamification, fitness activity, or walking on a virtual tour)? Additionally, what kind of activity could be helpful to gauge users’ biological responses? Is it the respiratory rate, blood pressure, or pulses under different techniques and approaches (control breathing technique vs. guided medication) that would be helpful?

The future direction of studies on the role of VR for psychological intervention for an individual with mental health issues should complement the existing body of knowledge that VR could be necessary for a more traditional style of face-to-face consultation settings [37,38,39,40,41]. VR may serve as a reachable and immersive way to bring practical clinical interventions to hospitalized patients, mainly during the ongoing COVID-19 pandemic across the world. It is pivotal to take note that from our systematic review findings, i.e., being immersive in the telepresence settings, engaging in an interactive process in a 3-dimensional format, engaging and activating an affective-motivational state by “escaping” to a virtual world are important elements to contemplate in VR software development for a psychological tool intervention.

### 4.6. Limitations of VR as a Digital Health Intervention: For Patients and the Healthcare Staff

Regarding the users who use the VR intervention such as healthcare staff, Kolbe et al. (2021) [29] outlined further limitations, including staff time pressure when medical and healthcare personnel are involved. This situation may pose a practical restriction on their utilization of the VR platform as a digital health intervention. VR is undoubtedly a unique context for patients via a self-directed experience rather than within a therapeutic session [42]. VR is affordable for many people, but not all patients are accessible through these treatment modalities [43]. For example, in a remote place with internet connection problems, the use of VR is minimal, or if the Wi-Fi connection is slow, the intervention could be inadequate.

### 4.7. Strength and Limitations of the Study

To the best of our knowledge, this is the first systematic review investigating the role of psychological intervention among individuals with mental health problems. The thoroughness of the findings is one of the major advantages of this study. There was a systematic literature search into all accessible resources, minimizing selection bias, and avoiding subjective selection bias. On the other hand, systematic reviews would provide with experts’ intuitive, experiential, and explicit perspectives on focused topics when narrated by experts in specific research areas [44]. In an earlier narrative review, the beneficial effects of VR were stated to promote positive well-being [44].

As with all other studies, the value of a systematic review may depend on what it accomplished in the research, i.e., what was observed, and the clarity of information reporting. The observation of the quality of systematic reviews varies based on the robustness of the included studies, which may limit the readers’ ability to assess the strengths and weaknesses of each inclusive study [45]. Even though systematic reviews are deemed to be the best evidence for making the best answer to a research question, there are certain intrinsic flaws related to them, such as the selection bias and heterogeneity of the included studies, i.e., diverse objectives outcomes (perceived self-satisfaction vs. relaxed state), different study settings (inpatient setting vs. online survey), different measurement tools (perceived self-rating assessment vs. qualitative thematic analysis), and different design platforms (2-D vs. 3-D), wherein a meta-analysis cannot be synthesized.

## 5. Conclusions

The COVID-19 pandemic has caused significant changes to most aspects of our lives. Because of the quarantine enforced by governments and authorities worldwide, people suddenly had to adapt their daily routines, including work, study, diet, leisure, and fitness activities, to the new circumstances. There is a necessity for the remote delivery of psychological and mental health interventions such as relaxation techniques, breathing exercises, and biofeedback.

This pandemic has inspired the need for creativity and substitution in medical and healthcare benefit delivery modalities. In addition, the following stages would seek to distinctly measure the comparative effects of diverse types of modules within the VR system, i.e., in vivo (traditional face-to-face) vs. virtual reality. Types of the frequency use (self-empowerment vs. default/regimented program), number of users (single vs. multiple players), the schema of the interaction (didactic coaching vs. free and easy trip, i.e., 360° virtual tours) are numbers of the practical troubleshooting that need to be solved. This step is essential to regulating patient satisfaction in a user-friendly, affordable, and medically feasible therapeutic modality. As we are not sure which mental health problems, i.e., mood disorders, anxiety, or psychosis maybe benefit from digital technology intervention, further studies are pivotal to measure specific mental health problem outcomes. We suggest embarking on a specific intervention for each mental health problem in order to evaluate and determine the effectiveness of the digital technology intervention.

## Figures and Tables

**Figure 1 ijerph-19-05389-f001:**
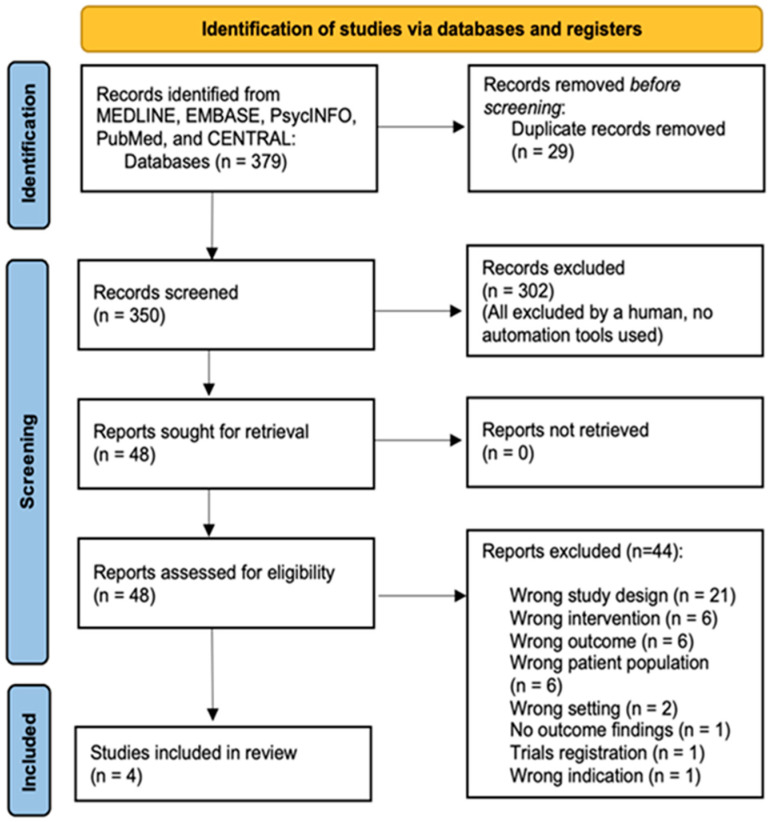
Systematic literature search, screening, and relevance assessment conducted according to PRISMA guidelines.

**Figure 2 ijerph-19-05389-f002:**
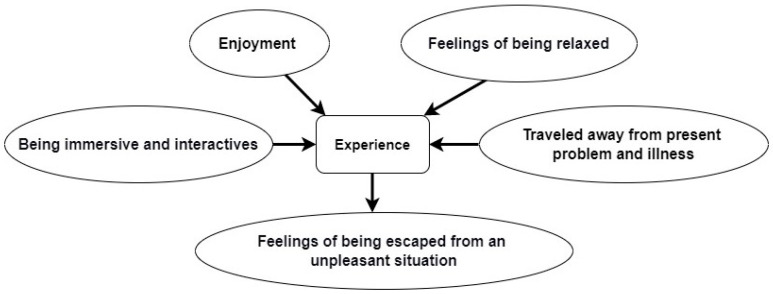
Examples of some patients’ suggestion in response to their VR experiences.

**Table 1 ijerph-19-05389-t001:** Characteristics of the included studies.

Study by Authors	Type of Study	Sample Size	Age	Country	Setting	Inclusion Criteria	Exclusion Criteria	Intervention	Exposure Measurement Scale	Outcome Measurement	Comparator/ Control	Statistics (e.g., OR/RR, *p*-Value, 95% CI)	VR-Based Intervention (Outcome)
Waller 2021 [27]	RCT	32	17–28	Canada	Setting was not defined	Not defined	Respondents were not controlled. The respondents underwent evaluation for life events, childhood events, traumatic events, PTSD, and life experiences before commencing the study	The non-VR group was exposed to (1) conventional face-to-face (in vivo (IV) method), (2) pre-recorded 360° video viewed by a standard laptop computer monitor (2D format), and (3) pre-recorded 360° video viewed through an HMD (VR condition; 3D format)	A revised evaluation of the emotional questionnaire, Buddhist Affective States, Meditation Breath Attention Scores, and Meditative Experience Questionnaire	Encountering relaxation, less distractibility from the process of breathing, and being less fatigue	3D (VR) vs. 2D format	Qualitative thematic analysis	↑ When compared to the 2D format, VR meditations were associated with a more significant outcome
Siani 2021 [28]	Cross-sectional	646	18–40	UK-based with multiple countries’ participation	Online survey on researcher’s personal Facebook and Twitter, Facebook group (Virtual Reality Society, Oculus Virtual Reality), and Reddit channels	Not defined	Not defined	Mainly VR-video games play	N/A	Increased use of VR during quarantine, to study the impact on mental health, devices type, and fitness intensity	Control (waiting list) vs. Two-Group Random Assignment Pretest–Posttest Design	Majority were positive about the usefulness of VR for fitness (χ2 = 185.21,df = 4, *p* < 0.001) and mental health (χ2 = 416.27, df = 4, *p* < 0.001). The majority of both VR (48.4%) and console (42.1%) users engaged with moderate intensity. A greater proportion of VR users engage in vigorous activity (43.0%) than mild activity (8.6%), a trend which is reversed in console users (38.0% mild, 19.8% vigorous)	↑
Kolbe 2021 [29]	Cross-sectional	24 (13 patients and 11 staff from COCID-19 Rehabilitation Unit (CRU), respectively	N/A	USA	COCID-19 Rehabilitation Unit (CRU)	(1) Hospitalized patients with +ve COVID-19 PCR test (2) Medical team deems the patient physically stable and has ongoing medical and rehabilitative needs (3) Able to tolerate >30+ min physical therapy (PT)/occupational therapy (OT) each daily (4) PT or OT recommendation for acute/subacute rehabilitation at the time of discharge (5) Anticipation of remaining in hospital/rehabilitation for ≥1 week	(1) Sexually not active, severe dementia and active delirium, or 1:1 sitter (2) must have non-invasive O2 needs of 6 L or fewer, or in case of tracheostomy patients have achieved “trach collaring” with anticipated ability to downsize/ decannulate	(1) Guided meditation, (2) exploration of natural environments, (3) cognitive stimulation game	A yes or no simple rating scale of 1–10 scores where 10 indicates the highest satisfaction and highest recommendation	Satisfaction, perceived enhancement	CRU inpatients and staff	For patients: 100% of patients answered “yes” to recommending the therapy to others, and 92.3% answered “yes” to the perceived enhancement of their treatment; For staff: 100% of staff answered “yes” to recommending the therapy to others, and 100% answered “yes” to perceived enhancement of their wellbeing	↑ The use of VR led to significant decreases in participants’ psychological stress
Yang 2021 [30]	Cross-sectional	235	>18	China	Local populace (in a shopping mall in Zhuhai City)	Not defined	Not defined	Validation of a theoretical model of the 360 degrees VR: A theoretical construct comprising the following factors: EN, IN, SA, SP, SR, TP are strongly related to each other and may help reduce stress from the COVID-19 pandemic	A newly designed questionnaire (translated and back-translated from English to Chinese) on the following features: (1) Introduction to the 360◦ virtual tours, and then the respondents were asked to watch a short video of the 360° virtual tours; (2) Measurement on the stress reduction of the research model; (3) Recorded the respondents’ demographic characteristics such as gender, age, marital status, education, income, occupation, and so on; (4) Measuring stress reduction as a result of using the 360° virtual tours	Enjoyment (EN) Involvement (IN) Satisfaction (SA) Sense of presence (SA) Stress Reduction (SR) Telepresence (TP) factors	No control group defined	PLS-SEM: The sense of presence (SP) and their level of enjoyment (EN) (β = 0.221, *t*-statistics (*t*) = 2.256), (SP) with involvement (IN) (β = 0.250, *t*-statistics = 3.224), SA and SP (β = 0.289, *t* = 4.099) TP to EN (β = 0.528, *t* = 5.411) TP to IN(β = 0.466, *t* = 6.028), TP to SA (β = 0.235, *t* = 3.246), path coefficients: EN to SA to stress reduction of COVID 0.268 (*t* = 4.345) and 0.474 (*t* = 5.904), respectively. 0.164 IN to stress reduction of COVID-19 (*t* = 2.626) and 0.158 (*t* = 2.093), SA to stress reduction from COVID-19, respectively: 0.196 (*t* = 3.116)	↑ Satisfaction with the 360° virtual tour experience and stress reduction

↑ = increased benefit and advantages; VR = virtual reality; OR = odds ratio; RR = relative risk; 95% CI = 95% confidence interval; β = Beta statistics; PLS-SEM = partial least squares structural equation modelling; LDT = letter-detection test.

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
