# Peer review of "Virtual Reality (VR) Technology for Treatment of Mental Health Problems during COVID-19: A Systematic Review"

_ijerph, 2022, doi:10.3390/ijerph19095389_

Round 1
Reviewer 1 Report
I think this study has failed to communicate new findings. Assume that the systematic literature review went well, as the authors claim. So, what is the conclusion of this paper? Is it a guideline? If the potential readers want to know about the role of VR technology during the pandemic, can they just read this paper?
Finally, it has to be pointed out that the number is too small for the analysis of four key papers. In the end, this study was limited to a summary of four papers. Can this be considered a systematic literature review?
Author Response
Reviewer.1
I think this study has failed to communicate new findings. Assume that the systematic literature review went well, as the authors claim.
We strictly followed the recommendations by the expert as evident in the systematic review guidelines stated in the methodology section, in p. 2: “This systematic review conformed to the Preferred Reporting Items for Systematic Reviews and Meta-analysis (PRISMA) guidelines [22,23]”.
So, what is the conclusion of this paper? Is it a guideline? If the potential readers want to know about the role of VR technology during the pandemic, can they just read this paper?
As the findings were based on 4 papers that we managed to extract using a guideline on how to report on the PRISMA guidelines, we concluded our findings in the final segment (as shown below) under the conclusion section. In the conclusion section, we concluded by saying:
“Our systematic review study revealed that VR is beneficial as a psychological tool for intervention for individuals with mental health problems. Being immersed in the telepresence, interacting in a 3-D format compared to a 2-D layout, having a sense of enjoyment and engagement, activating an affective-motivational state, "escaping" to a virtual from the real world are pivotal faucets of VR as a psychological tool for intervention. These elements – being immersive, interactive, and having a sense of engagement/enjoyment from the result of our study should be integrated or incorporated into the digital technology and intervention for a person with mental-health morbidity.”
Finally, it has to be pointed out that the number is too small for the analysis of four key papers. In the end, this study was limited to a summary of four papers. Can this be considered a systematic literature review?
Our study adhered to a systematic review process. The process of our review is comprehensive enough to establish consistency and generalizability of research findings across settings and populations. The research question has been explicitly stated, with clearly defined inclusion and exclusion criteria, and the search is exhaustive in nature with the inclusion of databases including references to databases in the computer study domain of the ACM digital library. Moreover, the protocol for this review was prospectively registered on PROSPERO to ensure transparency, scientific and good ethical practices.
We strictly followed the standards in the systematic review guidelines, i.e., PRISMA recommendation on how to report a systematic review and meta-analysis (stated in the methodology section).
Admittedly, there is no consensus on how many studies are required for systematic review. To proceed with meta-analysis, however, at least two studies are required. As the studies extracted were heterogeneous and diversified, i.e., each study has different objectives outcomes (perceived satisfaction vs. relaxed state), different settings (inpatients vs. online survey), different assessment tools (perceived self-rating vs. qualitative thematic analysis), and different design platforms (2-D vs. 3-D), we cannot synthesize the findings into a meta-analysis.
Reviewer 2 Report
General:
The topic of the current article is extremely useful as well the objective is relevant. Even so, I believe that from the methodological point of view, it has a few limitations, which I will enumerate below.
This work can be accepted with revision
Abstract:
- No specific comments
Introduction
- The introduction is very well organized, and the problem is very well presented.
Materials and methods
- This section is very well organized
- Regarding the Eligibility Criteria, it is not clear why the authors chose these criteria. The authors are advised why these criteria were chosen
- Concerning psychological intervention, the authors are advised to define the type of intervention
- Regarding the following sentence “The criteria for interventions/exposure include: psychological consultation and/or intervention delivered using VR in an inpatient or outpatient setting”, the authors should explain what kind of intervention is included. For example, it is included rehabilitation intervention.
Results
- The authors must explain the following sentence “44 studies were excluded due to the following reasons”. It is not clear the reasons. The authors must justify the reason
- Regarding table 1, the authors are advised to explain the choose of each variable. For example, it is not clear why country is important to mentioned
Discussion
- No specific comments .
Conclusion
- No specific comments
References
- No specific comments
Tables
- No specific comments
Author Response
Reviewer. 2
General:
The topic of the current article is extremely useful as well the objective is relevant. Even so, I believe that from the methodological point of view, it has a few limitations, which I will enumerate below.
This work can be accepted with revision
Abstract:
No specific comments.
We acknowledge the precious comments with due thanks.
Introduction
The introduction is very well organized, and the problem is very well presented.
We are grateful to the honorable reviewer for the comments.
Materials and methods
- This section is very well organized.
We thank the honorable reviewer for the comments.
- Regarding the Eligibility Criteria, it is not clear why the authors chose these criteria. The authors are advised why these criteria were chosen.
The criteria were chosen in order to include studies that investigated the utilization and/or usefulness of VR as a tool in psychological intervention in subjects with or without psychological distress.
We mentioned it on p. 4, first paragraph, under the subheading of 2.2. Eligibility criteria:
“The criteria for interventions/exposure include psychological interventions (e.g., counseling, family-based intervention, psychotherapy, behavioral therapy, and positive activity intervention (PAI)s) delivered using VR in an inpatient or outpatient setting.”
- Concerning psychological intervention, the authors are advised to define the type of intervention
We have included some psychological interventions in the conclusion section. In p. 14, first paragraph, we wrote:
“There is a necessity for the remote delivery of psychological and mental health interventions such as relaxation techniques, breathing exercises, and bio-feedback”.
- Regarding the following sentence “The criteria for interventions/exposure include: psychological consultation and/or intervention delivered using VR in an inpatient or outpatient setting”, the authors should explain what kind of intervention is included. For example, it is included rehabilitation intervention.
We have added accordingly. In the search strategy, we included all types of information, such as intervention, including rehabilitative intervention, in order to find the source of any relevant studies on this subject matter. To increase the chance of getting all necessary information, we used both the Boolean technique (AND, OR) and the truncation approach (*) as our search strategy. On p. 2, we added these search strategies and wrote:
“The search-related terms incorporating the Boolean and the truncation were as follows: (Virtual reality OR simulated-3D-environment OR VR) AND (covid! or corona!) AND (mental* OR psychologic* OR well* OR health*) AND (intervention).”
Results
The authors must explain the following sentence “44 studies were excluded due to the following reasons”. It is not clear the reasons. The authors must justify the reason
We have explained the reasons as we wrote in p.4 (under 3. Results subheading 3.1. Study characteristics section):
“A total of 44 studies were excluded due to the following reasons: wrong study design (21) (e.g. case reports), wrong intervention (6) (e.g. augmented reality [AR], face-to-face consultation), wrong outcomes (6) (e.g. education satisfaction), wrong patient population (5) (e.g. healthy or not related to mental health), wrong setting (2) (outside the time frame of the COVID-19 pandemic), no outcome findings (1) (e.g. research protocol without outcome results), trials registration (1), wrong indication (1), and commentators (1).”
Regarding table 1, the authors are advised to explain the choose of each variable. For example, it is not clear why country is important to mentioned
Thank you for your comments. Based on our reading and understanding (http://dx.doi.org/10.1136/ebnurs-2021-103417 on five tips for developing useful literature summary tables for writing review articles), the commonly used information includes authors, purpose, methods, key results, and quality scores. We think the country of origin of the study would add more information on the implementation of that particular intervention. While extracting all relevant information is important, such templates should be tailored to meet the needs of the individual’s review. For example, for a review of the effectiveness of healthcare interventions, a literature summary table must include information about the intervention, its type, content timing, duration, setting, and effectiveness, among others.
On p.4, before Table 1, we wrote: “We summarized the key results of the study characteristics, which include information about the intervention, its type, duration, setting, and effectiveness in Table 1.”
Discussion
- No specific comments.
Noted with thanks.
Conclusion
- No specific comments.
Noted with thanks.
References
- No specific comments.
Noted with thanks.
Tables
- No specific comments
Noted with thanks.
Reviewer 3 Report
Dear Editor,
I really appreciate the opportunity to review the manuscript ijerph-1657757 entitled:
"The Role of Virtual Reality (VR) as a Psychological Intervention Tool for Mental Health Problems During
Covid-19 Pandemic: A Systematic Review"
I commend the authors for describing this critical and timely issue. The paper is interesting and well-written; however, I would like to highlight some issues that merit revision:
Based on the title, the reader expects to get some indication, at least cursory, of what the application areas of these technologies might be. Although the article is very thorough and well documented these aspects should be emphasized in the conclusions, i.e., these technologies apply favorably especially to:
- Mood disorders?
- Anxiety?
- Psychosis?
I kindly ask the authors to describe this, or possibly add in the limitations because this was not possible to discuss.
Author Response
Dear Editor,
I really appreciate the opportunity to review the manuscript ijerph-1657757 entitled:
"The Role of Virtual Reality (VR) as a Psychological Intervention Tool for Mental Health Problems During Covid-19 Pandemic: A Systematic Review"
I commend the authors for describing this critical and timely issue. The paper is interesting and well-written; however, I would like to highlight some issues that merit revision:
Based on the title, the reader expects to get some indication, at least cursory, of what the application areas of these technologies might be. Although the article is very thorough and well documented these aspects should be emphasized in the conclusions, i.e., these technologies apply favorably especially to:
- Mood disorders?
- Anxiety?
- Psychosis?
To be honest, one cannot be sure which mental health problems (mood disorders, anxiety, or psychosis) maybe benefit from the digital technology intervention, as the studies which were included in this systematic review were very heterogeneous and diversified. None of the studies really measured specific mental health problem outcomes. We suggest embarking on a specific intervention for each mental health problem in order to evaluate and determine the effectiveness of the digital intervention.
We included this area of interest in the last paragraph of the text. On p. 15, we wrote:
“As we are not sure which mental health problems, i.e., mood disorders, anxiety, or psychosis maybe benefit from the digital technology intervention, further studies were pivotal to measure specific mental health problem outcomes. We suggest embarking on a specific intervention for each mental health problem to evaluate and determine the effectiveness of the digital technology intervention.”
I kindly ask the authors to describe this, or possibly add in the limitations because this was not possible to discuss.
We described the limitation of the systematic review. On p. 14, we wrote:
As with all other studies, the value of a systematic review may depend on what was accomplished in the research, i.e., what was observed, and the clarity of information reporting. The observation of the quality of systematic reviews varies based on the robustness of the included studies which may limit the readers’ ability to assess the strengths and weaknesses of each inclusive study [46]. Even though systematic reviews are deemed to be the best evidence for making the best answer to a research question, there are certain intrinsic flaws related to it, such as the selection bias and heterogeneity of the included studies, i.e., diverse objectives outcomes (perceived self-satisfaction vs. relaxed state), different study settings (inpatients setting vs. online survey), different measurement tools (perceived self-rating assessment vs. qualitative thematic analysis), and different design platforms (2-D vs. 3-D), where a meta-analysis cannot be synthesized.
We have added a new reference in the text in order to describe the limitation of a systematic review (in the reference section):
- Moher D, Liberati A, Tetzlaff J, Altman DG PRISMA Group. Preferred reporting items for systematic reviews and meta-analyses: The PRISMA statement. PLoS Med. 2009;6:e1000097